# Targeting the Redox Landscape in Cancer Therapy

**DOI:** 10.3390/cancers12071706

**Published:** 2020-06-27

**Authors:** Dilip Narayanan, Sana Ma, Dennis Özcelik

**Affiliations:** 1Department of Drug Design and Pharmacology, University of Copenhagen, Universitetsparken 2, 2100 Copenhagen, Denmark; dilip.narayanan@sund.ku.dk (D.N.); sana.ma@sund.ku.dk (S.M.); 2current address: Chemistry | Biology | Pharmacy Information Center, ETH Zürich, Vladimir-Prelog-Weg 10, 8093 Zürich, Switzerland

**Keywords:** oxidative stress response, reactive oxygen species, Nrf2–Keap1 signaling pathway, antioxidants, redox homeostasis, exosomes, extracellular vesicles, tumor redox microenvironment, hypoxia, drug development

## Abstract

Reactive oxygen species (ROS) are produced predominantly by the mitochondrial electron transport chain and by NADPH oxidases in peroxisomes and in the endoplasmic reticulum. The antioxidative defense counters overproduction of ROS with detoxifying enzymes and molecular scavengers, for instance, superoxide dismutase and glutathione, in order to restore redox homeostasis. Mutations in the redox landscape can induce carcinogenesis, whereas increased ROS production can perpetuate cancer development. Moreover, cancer cells can increase production of antioxidants, leading to resistance against chemo- or radiotherapy. Research has been developing pharmaceuticals to target the redox landscape in cancer. For instance, inhibition of key players in the redox landscape aims to modulate ROS production in order to prevent tumor development or to sensitize cancer cells in radiotherapy. Besides the redox landscape of a single cell, alternative strategies take aim at the multi-cellular level. Extracellular vesicles, such as exosomes, are crucial for the development of the hypoxic tumor microenvironment, and hence are explored as target and as drug delivery systems in cancer therapy. This review summarizes the current pharmaceutical and experimental interventions of the cancer redox landscape.

## 1. Introduction

Redox biology is a vastly complex and heterogeneous field that has drawn increasing attention in research due to its fundamental implications for our understanding of physiological function [1]. The concept of redox biology usually operates with a set of defined oxidants and antioxidants, which can lead to redox stress if the equilibrium of both classes of redox molecules becomes imbalanced [2,3]. In the case of archetypical oxidative stress, this balance is tilted towards the class of oxidants, which usually comprise reactive oxygen species (ROS) [4]. The cell aims to restore the redox balance by employing an antioxidant defense system. If the balance cannot be restored, the relative elevation of ROS levels leads to the development of diseases, including cancer [5,6,7]. 

The detailed role of ROS in cancer development remains unclear since the role of ROS varies greatly among different cancer types, tissues, and stages [8,9]. The general consensus in research suggests that high ROS production induces carcinogenesis by impairing the DNA repair system, which subsequently leads to an accumulation of DNA damage, such as base modifications, inter- and intrastrand bindings and DNA–protein crosslinks [10]. In addition, increased H_2_O_2_ and O_2_^•−^ is associated with increased cancer cell proliferation [11]. Cancer cells show an altered metabolism, which demonstrates a substantially increased production of ROS [12,13,14]. A comprehensive review of the metabolic regulation of redox balance in cancer was published earlier by Purohit et al. [15]. With continuous growth, cancer cells consolidate into a tumor that faces cycles of hypoxia and reoxygenation [16]. Hypoxic conditions stimulate remodeling of the tissue microenvironment in order to ensure influx of nutrition, efflux of waste products, and establishment of a suitable redox microenvironment [12]. Adaption to hypoxia is a crucial step in the transformation of a cell to a malignant state [17]. This process is accompanied by blood vessel development, which is often devoid of coordinated structure and organization, and induces oxidative stress through periods of changing redox environment [16,18]. Studies suggest that cancer cells engage in intercellular communication with the tumor and the surrounding tissue via extracellular vesicles (EVs), such as exosomes, in order to create a metabolic microenvironment that fosters tumor growth and metastasis [19].

Since redox biology is linked tightly to cancer, current pharmaceutical developments include the design of compounds that target key players and processes within the oxidative and antioxidative landscape in the cancer cell. For instance, chemo- and radiotherapy are employed to induce overproduction of ROS and, hence, apoptosis of tumor cells [20]. The application of such interventions, however, depends on the genetic polymorphism of the patient, on cancer type and stage, and on the affected tissue [8,21]. Other approaches aim to utilize redox-driven strategies to modulate immunotherapy in cancer therapy [22,23]. In this review, we provide an overview of the major components of both the oxidative and the antioxidative landscape and their connection to the development of cancer drugs. In addition, we present examples of current efforts that aim to modulate key proteins of the redox landscape in cancer, which is summarized in Table 1.

## 2. The Oxidative Landscape in Cancer

A cornerstone of the cellular redox landscape is the interplay of three organelles: mitochondria, the endoplasmic reticulum (ER), and peroxisomes [95]. The contribution of mitochondria, peroxisomes and the ER to the intracellular production of ROS varies among cells, tissues, and the general redox environment. Studies on perfused liver tissue indicate that peroxisomes produce the largest absolute amount of ROS [96]. Mitochondria can contribute substantially to general ROS production as well [97]. In comparison however, the ER provides the highest relative amount of cytosolic ROS due to the lack of antioxidative systems in the ER [95]. The predominant sources of ROS in both normal and cancer cells comprise NADPH oxidases (NOXs) and the electron transport chains (ETC) in the mitochondria, whereas the ER can also serve as a substantial source of ROS due to ER oxidoreductases and NOXs [15,95,98,99]. Both NOXs and mitochondrial ETC reduce oxygen to the highly reactive superoxide anion (O_2_^•−^). O_2_^•−^ subsequently undergoes a complex series of conversion reactions, giving rise to more stable hydrogen peroxide (H_2_O_2_) but also to more toxic ROS, e.g., hydroxyl radical (^−^OH), or reactive nitrogen species (RNS), e.g., nitric oxide (NO^−^). An overview of the cellular oxidative landscape is presented in Figure 1. Since increased ROS production is associated with cancer development, pharmaceutical research aims to modulate the oxidative landscape in cancer therapy.

### 2.1. The Mitochondrial Electron Transport Chain

The ETC in the inner mitochondrial membrane comprises four multi-enzyme complexes, termed complexes I, II, III, and IV. They drive an electrochemical proton gradient across the membrane, which can be used for ATP generation via ATP synthase (also called complex V) or heat generation via protein uncoupling. Electrons, leaked from the ETC at complexes I, II and III, are the major source of O_2_^•−^ and other ROS in the mitochondria [100,101]. Mutations in the ETC complexes can disturb the electron chain reaction, leading to elevated ROS levels, and hence contribute to cancer proliferation and metastasis [102]. Hence, ETC complex inhibitors are actively pursued in drug discovery and development of novel anticancer drugs [103].

Mutations in complex I, the largest of all complexes, occur frequently in many different tumors and are considered essential for the glycolytic switch, known as the Warburg effect [104], and for ROS-driven metastasis [105,106]. However, some mutations appear to have tumor-suppressor effects, suggesting that pharmaceutical targeting needs to consider type and stage of the cancer prior to the start of the intervention [107]. As of today, a number of approved complex I inhibitors are available, including the well-known and established diabetic drugs canagliflozin [27,28] and metformin [30], which are also being investigated as anticancer drugs. In addition, some recently approved compounds, e.g., celastrol [29], BAY 87-2243 [24,25,26], and xanthohumol [32,33], also target complex I, resulting in mitochondrial ROS production and anticancer effects.

Complex II, the smallest of the respiratory complexes, has also drawn considerable attention since it is positioned at the intersection between the ETC and the TCA cycle [37]. Many compounds have been identified as potent complex II inhibitors, but clinical application is hampered by the high degree of toxicity, for instance, in the cases of troglitazone and thenoyltrifluoroacetone. Nonetheless, vitamin E analogues, such as tocopherols and tocotrienols, have shown promising results in preclinical trials [38,39]. In addition, 3-Bromopyruvate derivatives have also produced promising results in preclinical trials and have advanced to clinical trials [34]. Notably, the anticancer drug lonidamine (LND) has been reported to inhibit complex II, and to increase the overall treatment response in cancer patients in combination with standard-of-care drugs, e.g., doxorubicin [35,36]. However, LND showed limited efficiency in clinical phase 3 trials but was recently modified into mito-lonidamine (Mito-LND), which is 100-fold more potent in cell culture and mouse models, and inhibits complex I and II [31].

Complex III can also contribute to cancer development. Mutations in this complex are associated with increased ROS production and apoptotic resistance, which is linked to accelerated growth in cancer cells [108]. A study showed that the long-established antimalarial drug, atovaquone, targets complex III and has anticancer properties [40]. This eventually led to two clinical trials, one of which investigated the effect of atovaquone on the tumor microenvironment of solid tumors [Atovaquone as a Tumour hypOxia Modifier (ATOM), NCT02628080] and the other on the outcome of chemotherapy in acute myeloid leukemia [Atovaquone (Mepron®) Combined With Conventional Chemotherapy for de Novo Acute Myeloid Leukemia (AML) (ATACC AML), NCT03568994].

The last of the four ETC complexes is the copper-dependent complex IV (or cytochrome c oxidase), which is the rate-determining enzyme of the ETC and crucial for cellular energy production [109]. It has been shown that tumors often need more copper [110] and that the copper chelator bis-choline tetrathiomolybdate (also known as SOD1 inhibitor ATN-224) inhibits complex IV activity in cancer cells [43]. Taken together, this indicates that inhibitors of ETC complex IV possess intriguing translational potential in the treatment of cancer.

### 2.2. ROS-Generating Enzymes of the Mitochondria

In addition to the ETC, mitochondria harbor enzymes that are also a source of ROS. One notable example at the inner mitochondrial membrane is dihydroorotate dehydrogenase (DHODH). DHODH is associated with complex III, and generates O_2_^•−^ and H_2_O_2_ [44]. Interestingly, DHODH was unknown until researchers identified the target of the anticancer drug brequinar [111]. Despite successful initial clinical trials, brequinar eventually failed due to inconsistencies in patient response [44]. Recent studies, however, show beneficial effects for administration of brequinar in chemotherapy, suggesting that brequinar can still be used in combination therapy for treatment of cancer patients [112]. Another established DHODH inhibitor is teriflunomide, which is approved for the treatment of multiple sclerosis [113]. Furthermore, teriflunomide, the prodrug of leflunomide, is approved for the treatment of rheumatoid and psoriatic arthritis [45]. Both drugs have been evaluated for various diseases and showed anticancer properties [44,46,47,48]; however, these compounds causes considerable adverse reactions due to significant off-target effects. This led to the termination of a clinical trial in which melanoma cancer was treated with a combination of leflunomide and the anticancer drug vemurafenib [Leflunomide+Vemurafenib in V600 Mutant Met. Melanoma, NCT01611675]. Despite these setbacks, leflunomide and teriflunomide remain the only FDA-approved DHODH inhibitors for clinical application, and are currently explored for other types of cancers [44]. 

Another ROS-producing enzyme in the inner mitochondrial membrane is glycerol-3-phosphate dehydrogenase 2 (mGPDH or GPDH2), which produces ROS through reverse electron transport from flavin adenine dinucleotide (FAD) to the ETC [114]. Much of mGPDH’s function remains to be established, but recent findings demonstrated that mGDPH inhibition impedes prostate cancer, indicating a pharmaceutically relevant role of this enzyme in cancer development [115]. Nevertheless, available inhibitors lack selectivity, and reports of two potent and selective mGPDH inhibitors, i.e., iGP-1 and iGP-5, were never followed up [49].

In contrast to DHODH and mGPDH, mitochondrial monoamine oxidase (MAO) is located at the outer membrane. Human cells express two variants, MAO-A and MAO-B, which contribute to ROS production in mitochondria by oxidative deamination of serotonins or catecholamines [116]. MAO-A is found in several tissues, for example, in the prostate, whereas MAO-B expression is limited to platelets; however, both variants are also expressed in the brain and contribute to neurological disorders [117]. While MAO-B is currently explored as a drug target in various neuropathologies [118], only MAO-A has been associated with cancer. For instance, MAO-A is overexpressed in prostate cancer and contributes to tumorigenesis [51]. Inhibitors for MAO-A have been in use since the 1950s for the treatment of major depression [119], but they are now studied as anticancer drugs as well. Phenelzine, a notable example of a potent non-selective and irreversible MAO-A inhibitor, is currently under investigation in clinical trials for the treatment of prostate cancer [50] [Phenelzine Sulfate in Treating Patients With Non-metastatic Recurrent Prostate Cancer, NCT02217709].

### 2.3. The Endoplasmic Reticulum

The ER is an intracellular network of membranes that is involved in a variety of basic physiological processes: in particular, protein synthesis, posttranslational processing, protein folding and transportation, as well as Ca^2+^ signaling and bioenergetics at mitochondria–ER contact sites [120]. The ER is an important player in the redox environment of the cell, which comprises two major sources of ROS. One of the sources of ROS is NOX4, a member of NADPH oxidase family, which we will discuss subsequently. The second major source of ROS is the Ero1α–PDI protein folding pathway [121]. One component of this pathway is protein disulfide isomerase (PDI). PDI, an abundant protein in the ER, is the founding member of a family of 20 related proteins in the ER, and essential for protein folding in the ER [122,123]. PDI-family proteins share at least one thioredoxin-like domain that contains a catalytic cysteine pair as part of a canonical CXXC motif [124]. The catalytically active domains of PDI catalyze a thiol-disulfide exchange reaction with cysteines of nascent client proteins, resulting in breakage, formation or isomerization of disulfide bonds [125,126]. This oxidoreductase-catalyzed reaction also requires the activity of the membrane-bound ER oxidoreductin 1α (Ero1α), a FAD-dependent oxidoreductase [127,128,129,130]. A byproduct of the Ero1α–PDI oxidative protein folding pathway is H_2_O_2,_ which contributes to a slightly oxidative environment within the ER, characterized by a high GSSG:GSH ratio [131]. Excess H_2_O_2_ is compensated by peroxiredoxin 4 (PRDX4), a member of the peroxiredoxin protein family, which is described in a later chapter [132]. Alternative models of oxidative protein folding involve quiescin–sulfhydryl oxidase 1 (QSOX1), which produces H_2_O_2_ and facilitates the formation of disulfide bridges in client proteins independently of Ero1α–PDI activity [133].

PDI upregulation correlates with cancer metastasis and invasion in various cancer types, and has drawn a lot of attention as a drug target in cancer therapy [134,135,136]. As of today, a large and growing number of PDI inhibitors have been discovered and characterized, such as RB-11-ca [67], 16F16 [56], origamicin [62,63], the phenyl vinyl sulfonate compound P1 [64], juniferdin [60], quercetin-3-rutinoside [66], ML359 [61] and PACMA31 [65], but none of them have progressed beyond preclinical studies. Nonetheless, two recent PDI inhibitors, CCF642 [57] and E64FC26 [58], demonstrated favorable results in preclinical studies in terms of potency, selectivity, and anticancer effects, and, hence, are promising candidates for clinical translation. Notably, isoquercetin is a PDI inhibitor that advanced the furthest in clinical use, and entered phase 2 trials in cancer patients a few years ago (Cancer Associated Thrombosis and Isoquercetin (CATIQ), NCT02195232). However, its primary application is not aimed at targeting cancer but at the inhibition of PDI activity in platelets in order to reduce the risk of thrombosis [59,137]. In contrast to this large pool of compounds targeting PDI inhibitors, only two inhibitors of Ero1α have been identified in a screen, i.e., EN460 and QM295, and no subsequent study was reported [55]. A lot of research is still needed, but recent developments of PDI inhibitors, as outlined above, indicate the potential of targeting the Ero1α–PDI protein folding pathway in cancer. 

### 2.4. Peroxisomes

Peroxisomes, formerly known as “microbodies”, are small organelles with a single membrane located in the cytoplasm of almost all eukaryotic cells [138]. The biogenesis of peroxisomes is still under debate as one model suggests growth and division whereas another model promotes de novo synthesis; regardless, both models agree on the contribution of the ER to the compartmentalization of the organelle [139,140]. A panel of transcription factors that are termed peroxisome proliferator-activated receptors (PPARs) regulate proliferation of peroxisomes [141]. Among many biological functions, peroxisomes are crucial for lipid homeostasis and cellular ROS metabolism [142]. Peroxisomes contain several flavin-dependent oxidoreductases, most notably xanthine oxidase (XO), which generates ROS [143]. In addition, peroxisomes also contain nitric oxide synthase (NOS2), which generates NO. To counterbalance ROS production, peroxisomes possess several antioxidant enzymes, such as catalase (CAT), superoxide dismutase 1 (SOD1), peroxiredoxin 5 (PRDX5), glutathione S-transferase kappa 1 (GSTK1) and glutathione peroxidase (GPx) [144,145]. The role of peroxisomes within the cellular redox landscape is not fully understood, but it has been suggested that their function as a source or sink for H_2_O_2_ is tissue-specific [144]. Nevertheless, peroxisomal function and redox metabolism are important for metabolic reprogramming and are crucially involved in cancer development [146]. In fact, various cancer types show decreased peroxisome levels, which is associated with overexpression of the negative regulator of peroxisome abundance and metabolism, termed hypoxia-inducible factor 2a (HIF2α) [147,148]. Current research efforts aim at peroxisomal ROS production, for instance, by targeting XO [149]. Inhibitors of XO, such as the purine analogue allopurinol, and the non-purine analogues febuxostat and topiroxostat, are approved for the treatment of gout and hyperuricemia, indicating that XO is a suitable therapeutic target [68,150]. Furthermore, there are several compounds that target PPARs in the treatment of cancer, demonstrating that modulation of peroxisomal function is a promising approach in disease management [151].

### 2.5. NADPH Oxidases 

NADPH oxidases (NOXs) play a key role in a wide range of physiological processes, such as gene expression regulation, cell signaling and differentiation. They are also crucially involved in many pathological processes, including cancer. The seven human NOX isoforms (NOX1 to NOX5, and the dual oxidases DUOX1 and DUOX2) are transmembrane proteins that transport electrons across the cytoplasm via FAD or across the extracellular membrane, using two heme groups, in order to generate O_2_^•−^ or H_2_O_2_ [152]. NOXs require various membrane and cytosolic protein subunits for their activity. For instance, the stability and activation of NOX1 to NOX4 and DUOX1 and DUOX2 depend on the stabilization partner p22phox and the maturation partners DUOXA1 and DUOXA2. The cytosolic subunits p47phox and NOXO1, and the activator subunits p67phox and NOXA1, are essential for NOX1 and NOX2 function [152,153]. NOX2, in particular, is an important member of the NADPH oxidase family. NOX2 was originally discovered in phagocytes as a source of ROS, which are employed in the defense against bacterial infection [153]. p47phox, the organizing component of NOX2, is phosphorylated and then translocated to the plasma membrane by p67phox, which activates the NOX2 complex [153]. 

Several studies show that cancer cells accumulate mutations, which increase ROS generation from NOX enzymes, eventually inducing tumorigenesis [153,154]. A particular type of mutation involves the GTPase KRAS, a member of the Ras oncogene family. KRAS mutations affect phosphorylation and membrane translocation of p47phox, and, thus, induce NOX1-mediated ROS formation and metastasis [154,155]. For instance, KRAS mutation increases NOX1 expression in colon adenocarcinoma and lung cancer cells [156]. 

Other NOX isoforms are also involved in cancer development. For example, upregulation of NOX4, the major ER NADPH oxidase, plays a key role in ovarian, pancreatic, kidney, and glioblastoma cells [157,158,159,160]. Recent in vivo studies reported that the inhibition of NOXs hinders tumor growth, indicating the pharmaceutical relevance of NOXs; however, current NOX inhibitors lack selectivity among NOX isoforms [161,162]. For instance, the natural organic compound apocynin inhibits p47phox membrane translocation, and thus activates NOX2, but it also inhibits Rho kinases, thus leading to cell cycle arrest [69,70,108,163]. In a recent study, Solbak et al. used fragment-based drug discovery to develop dimeric NOX2 inhibitors that target p47phox–p22phox protein–protein interaction [164]. The pan-NOX inhibitor VAS2870 interferes with NOX binding proteins, and hence inhibits NOX complex formation [54,165,166]. The two pyrazolopyridine compounds GKT136901 and GKT137831 show 10-fold selectivity towards NOX1 and NOX4 over NOX2 [52,53,167,168]. Notably, the NOX1/NOX4 dual inhibitor GKT137831 is the only NOX inhibitor that has entered clinical trials [GKT137831 in IPF Patients with Idiopathic Pulmonary Fibrosis (GKT137831), NCT03865927]. Nevertheless, recent clinical data failed to reproduce any pharmacological effects in humans, causing a decline in interest in pursuing NOXs as drug targets [169].

## 3. The Antioxidative Landscape in Cancer

The antioxidative defense plays a crucial role in maintaining an adequate redox environment for physiological cell function and survival in oxidative stress. For example, the production of O_2_^•−^ in mitochondria, peroxisomes and the ER is countered by different types of superoxide dismutases (SODs) that catalyze the disproportionation of O_2_^•−^ to H_2_O_2_ and O_2_. Eventually, H_2_O_2_ disproportionates to water and O_2_, completing the detoxification process of O_2_^•−^. Detoxification of H_2_O_2_ is thoroughly facilitated by various mechanisms, comprising the activity of the enzyme catalase (CAT) and oxidation of cysteine residues either in glutathione (GSH) or in peroxiredoxins (PRDXs). All of these antioxidative processes are under tight regulation by transcription factors and are upregulated during the oxidative stress response. One important transcription factor is Nrf2, which in turn is under the control of the negative regulator Keap1. An overview of the cellular antioxidative landscape is presented in Figure 2.

### 3.1. The Nrf2–Keap1 Signaling Pathway

The transcription factor, nuclear factor erythroid 2-related factor 2 (Nrf2), plays a central role in the antioxidative landscape. Upon oxidative stress, Nrf2 translocates to the nucleus and induces antioxidant response elements (ARE), a large array of various antioxidative factors that comprises antioxidative and cytoprotective enzymes, e.g., NQO1, GSH and thioredoxin [170] (Figure 2). Under basal conditions, however, the repressor protein Kelch-like ECH-associated protein 1 (Keap1) promotes polyubiquitinylation and subsequent proteasomal degradation of Nrf2, thus maintaining a low cellular concentration of Nrf2 [171].

The exact role of Nrf2 in carcinogenesis remains unclear. On the one hand, it has been shown that Nrf2 is upregulated in various cancers [172,173], which is caused either by DNA methylation in the promoter region of Keap1, constitutive Nrf2 activation, or mutations in the Keap1 domain [174,175]. Furthermore, basal Nrf2 levels can increase during chemo- or radiotherapy, which correlates with therapy resistance [173]. On the other hand, Nrf2 plays a protective role and prevents cancer development by reducing ROS levels [172,173]. This implies a dual role for Nrf2 in cancer development and suggests that optimal therapy likely depend on cancer stage or cancer type, as summarized in Milkovic et al. [176]. Consequently, there are two pharmaceutical approaches for targeting the Nrf2–Keap1 signaling pathway in cancer cells. 

One approach employs Nrf2 inhibitors in order to counter the effects of Nrf2 upregulation and reduce the oxidative stress response [71]. As of today, many different Nrf2 inhibitors have been described, for instance, the flavonoid luteolin, and some synthetic compounds, e.g., AEM1 and ML385, have shown promising results in cell lines; however, none are in clinical trials. An overview of available Nrf2 inhibitors can be found here [71]. 

The other pharmaceutical approach targets the cytosolic protein–protein interaction between Nrf2 and Keap1 in order to activate Nrf2 and to boost the oxidative stress response [173,177,178]. An earlier study demonstrated Nrf2 activation by genetic knockout of Keap1 in vivo as well as inhibition of the Nrf2–Keap1 interaction with covalent electrophilic modifiers like dimethyl fumarate (DMF) or peptides [179]. Due to insufficient specificity of these covalent modifiers and low bioavailability and cell permeability of the employed peptides, current efforts focus on alternative classes of compounds. Recent studies show successful induction of Nrf2 through targeting of the Nrf2–Keap1 protein–protein interaction with non-covalently interacting small molecules [180]. One example is the synthetic oleanane triterpenoid compound RTA 405, which showed antitumor activity in cell culture [76,77,181]. There are also a number of natural compounds, for instance, sulforaphane (SFN) and curcumin, which act as Nrf2 activators and show anticancer effects [72,75,78,177,178]. A comprehensive overview of current modulators of Nrf2–Keap1 protein–protein interaction is presented by Robledinos-Anton et al. [178]. In summary, several Nrf2 activators and inhibitors are in development and in different stages of clinical trials, but, so far, the only Nrf2 modulator in the clinic is DMF, which is approved for the treatment of multiple sclerosis and psoriasis [73,74,75,177,178].

### 3.2. Glutathione Homeostasis

Glutathione (GSH) is a ubiquitous antioxidant and the most abundant thiol in animal cells, with a local concentration of up to 10 mM [182]. GSH also occurs as an oxidized glutathione disulfide (GSSG) in the cytosol and organelles; hence, the GSSG:GSH ratio is an indicator of the cellular redox state [183]. GSH is synthesized by a two-step reaction (Figure 2): (1) glutamate cysteine ligase (GCL) conjugates the amino acids glutamate and cysteine to γ-glutamyl cysteine, followed by (2) the addition of glycine to the cysteine carboxyl by glutathione synthetase (GSS) [182]. 

Under normal conditions, the overwhelming majority of the cellular GSH pool is present in the reduced form, but during oxidative stress, the ratio shifts towards GSSG. In response to oxidative stress, cancer cells upregulate the GSH level, which correlates with cancer progression and resistance towards chemotherapy [184]. As of today, attempts to modulate the GSH pool in cancer have failed due to insufficient selectivity of the available compounds [185]. Current strategies include inhibition of GSH synthesis by targeting GCL or by interfering with uptake of cystine, the oxidized version of cysteine, through inhibition of the X_C_^−^ antiporter system [186]. Notably, buthionine sulfoximine (BSO) is a GCL inhibitor that has been shown to decrease the GSH level in cancer cells but failed to deliver translatable clinical benefits [79,187]; however, recent studies attempt to identify sensitive patients and cancer types for treatment with BSO [188].

Alleviating the effects of oxidative stress can be achieved by increasing the GSH level but also by reducing GSSG to GSH. This reaction is facilitated by glutathione reductase (glutathione–disulfide reductase; GSR) (Figure 2). GSR uses an FAD prosthetic group to transfer the reductive equivalent of NADPH to GSSG [189]. One study showed that GSR is associated with decreased ROS levels and anticancer drug resistance in glioblastoma cells, indicating a novel drug target [190].

In contrast, the oxidation of the cysteine thiol in GSH with H_2_O_2_ to GSSG is catalyzed by glutathione peroxidases (GPx). The GPx family, which has been summarized comprehensively by Brigelius-Flohe et al. [191], plays a crucial role in the protection against oxidative stress. There are eight human GPx isoforms that contain either a selenocysteine (GPx1-4 and GPx6) or a cysteine (GPx5, GPx7 and GPx8) as the active residue. All GPx isoforms vary in location and biological function. GPx1, located in the cytoplasm, is the most abundant isoform and uses mainly H_2_O_2_ as the substrate [192]. GPx4 is predominantly located in mitochondria and has a high affinity for lipid hydroperoxides [193], whereas GPx7 and GPx8 play an important role in the ER [194]. A number of published studies show the involvement of GPx proteins in tumorigenesis and chemotherapy resistance, which is summarized in these reviews [195,196]. Currently, there is no selective inhibitor for GPx proteins for therapeutic application; however, recent developments in medicinal chemistry show promising advancements for targeting GPx1 [197] and GPx4 [198]. 

In addition to its role as an ROS scavenger, GSH is involved in cellular detoxification processes. The diverse family of glutathione S-transferases (GSTs) conjugates GSH to biological substrates, e.g., xenobiotics or lipid peroxides, in order to promote further processing or excretion [199]. Lipid peroxides are often generated in peroxisomes and are detoxified by GSTK1, which is located in mitochondria, peroxisomes and the ER [200]. Notably, GSTs detoxify anticancer drugs in cancer cells and, according to several studies, GSTs play additional roles in cancer development, particularly glutathione S-transferase pi (GSTP) [201,202]. However, low selectivity has hindered the translation of compounds to a clinical setting [203].

### 3.3. The Peroxiredoxin–Thioredoxin System

Another important branch of thiol metabolism involves the highly conserved family of peroxiredoxins (PRDXs). PRDXs are key players in the antioxidant system because they play an important role in the detoxification of H_2_O_2_. There are six PRDX isoforms (PRDX1 to PRDX6) in the human genome that are abundantly expressed, highlighting their importance for redox balance and signaling [204,205]. The isoforms are located in different compartments of the cell. For instance, PRDX4 is found in lysosomes and the ER, whereas PRDX5 is also present in the mitochondria. Notably, PRDX6 is the only isoform that has been found in the extracellular environment. The active site of PRDX proteins contains a redox-active cysteine, known as the peroxidatic cysteine, which oxidizes to form sulfenic acid or engages in disulfide bond formation upon conversion of H_2_O_2_ to water. Overoxidation of the peroxidatic cysteine yields sulfinic or irreversible sulfonic acid, rendering PRDX inactive [206] (Figure 2). Sulfinic acid in PRDX can be reduced to sulfenic acid by sulfiredoxin, an antioxidative enzyme that is explored as a potential drug target in cancer therapy [207,208,209]. Reversible sulfenic acid in PRDXs is reduced by the thiol-containing thioredoxin, which exists as a cytosolic (TXN1) and a mitochondrial version (TXN2). It also maintains the redox state of its interaction partners [210,211]. Oxidized thioredoxin itself is reduced by the selenocysteine-containing active sites of the FAD- and NADPH-dependent thioredoxin reductases (TrxR1, TrxR2 and TrxR3), which play a central role in the thioredoxin system and in cell survival and DNA replication [212,213] (Figure 2). 

Multiple reports show upregulation of PRDXs in cancer and involvement in resistance to radiation therapy [214,215]. Currently, no approved inhibitors of PRDXs are available, but there are ongoing research efforts in the development of several compounds in preclinical development. One notable example is adenanthin, a natural diterpenoid that exhibits potent anticancer effects [216]; however, it was shown that adenanthin targets several redox pathways, and is hence not a selective PRDX inhibitor [217]. Another compound, AMRI-59, is a derivative of the natural antibiotic thiostrepton, which targets PRDX1 in cancer cells and shows radiosensitizing effects in cell culture [80,81]. 

Similar to PRDXs, a large number of studies have shown that thioredoxin and TrxR1 are overexpressed in various cancers and are associated with resistance to anticancer drugs [82,218,219,220]. Consequently, substantial drug development efforts target the thioredoxin system, in particular TrxR. These endeavors have yielded a large pool of TrxR inhibitors, including gold- or selenium-containing compounds, nitroaromatic compounds, polyphenolic compounds like curcumin derivatives, and, notably, the standard-of-care compounds cisplatin and arsenic trioxide (ATO) [84,85]. An overview of TrX inhibitors is presented in a review from Urig and Becker [86]. Other compounds are used for treatment of different diseases but are currently studied as anticancer drugs, e.g., the antirheumatic agent drug auranofin [221].

In addition to TrxR, researchers have developed three different types of inhibitors of thioredoxin. One compound is the small molecule inhibitor 1-Methylpropyl 2-imidazolyl disulfide (PX-12), which advanced into phase 1 clinical trials against solid tumors and phase 2 clinical trials against pancreatic cancer, but PX-12 eventually failed to deliver sufficient results [222,223]. The second compound 4-Benzothiazole-substituted quinol (PMX464) also showed anticancer properties but never advanced to clinical trials [83]. One report demonstrated efforts to repurpose both thioredoxin inhibitors, PX-12 and PMX464, as antiplatelet agents [224]. The third compound is the histone deacetylase inhibitor suberoylanilide, commonly known as Vorinostat (Zolinza). This first-in-class anticancer drug was approved by the FDA for the treatment of cutaneous T-cell lymphoma [225], and is currently under investigation in numerous clinical trials against many different types of cancer [82,218].

### 3.4. Superoxide Dismutase

A major player in the detoxification of ROS is the ubiquitous metalloenzyme SOD [226]. SOD catalyzes a two-step reaction converting two molecules of O_2_^•−^ into one molecule of O2 and one molecule of H_2_O_2_ [227] (Figure 2). The human genome encodes several types of SODs, which are all strictly compartmentalized [228]. The dimeric copper zinc SOD (CuZnSOD, SOD1) is located in the cytosol, nucleus, peroxisomes, and intermembrane space of the mitochondria [229]. The tetrameric manganese SOD (MnSOD, SOD2) is present in mitochondria and executes important functions in cell signaling [230]. The extracellular SOD (EcSOD, SOD3) is cell-type specific and mainly secreted in the cardiovascular endothelium, lungs, and placenta [231]. It also modulates the redox state of the extracellular environment. EcSOD contains a heparin-binding domain (HBD) enabling binding to heparin sulfate proteoglycans on the cell surface and the extracellular matrix [232]. 

The contribution of the individual SOD isoforms to cancer development is not fully understood, as summarized in a review by Che et al. [233]. The review describes that SOD1 is a known disease-causing gene, whereas the role of SOD2 is less clear, but the general consensus is that overexpression of SOD2 correlates with invasive and metastatic cancer. The contribution of EcSOD to cancer development is even less clear, but a growing body of research suggests that EcSOD is pro-oncogenic [233]. One study demonstrated that overexpression of EcSOD mediates tumorigenesis through modulation of the tumor microenvironment (TME) [234]. As of today, there are only a few SOD inhibitors available, and they all target SOD1. The most advanced SOD1 inhibitor is the copper chelator ATN-224, which has also been identified as an inhibitor of ETC complex IV (as mentioned above). Two clinical trials were launched to examine ATN-224 in solid tumors and in prostate cancer [41,42], but ATN-224 did not match the expectations set by preclinical studies and failed to show clinical significance. Another SOD1 inhibitor is the estrogen derivative 2-methoxyoestradiol (2-ME), which induces ROS production and selectively kills human leukemia cells while sparing normal lymphocytes [235]; however, 2-ME does not bind SOD1, as initially suggested, but interfered with the assay read-out [236]. The SOD1 inhibitor LCS-1 (4,5-Dichloro-2-m-tolylpyridazin-3(2H)-one) was discovered in a high-throughput screen, but no follow-up studies were reported to date [88]. A different approach is the use of SOD mimics during and after cancer radiotherapy in order to increase ROS detoxification and to mitigate damage of healthy tissue [237].

### 3.5. Catalase

The enzyme CAT is present in almost all cells exposed to oxygen and catalyzes the detoxification step of H_2_O_2_ [238]. CAT converts H_2_O_2_ to water and shows one of the highest turnover numbers of any known enzyme [239] (Figure 2). Human CAT contains four iron-containing heme groups and is mainly located in peroxisomes but is also present in the cytoplasm [240]. In cancer cells, CAT is often found in high concentrations at the plasma membrane [144,241] and sometimes released in the extracellular matrix [242,243,244]. There is conflicting data on the overall intracellular CAT concentration in cancer cells, likely due to tissue-specific effects [245]; nonetheless, it has been reported that CAT upregulation in cancer cells impairs chemotherapy [246]. Currently, there are several approaches to inhibit CAT in cancer therapy, aiming to elevate cellular ROS levels and thus inducing apoptosis in cancer cells. Current approaches focus on targeting membrane-associated CAT in cancer cells using antibodies [247] or exogenous singlet oxygen [248,249]. A recent study suggested that ATO causes down-regulation of CAT, indicating that CAT is a suitable anticancer drug [87,250].

### 3.6. NADPH Dehydrogenases (Quinone)

The cytosolic NAD(P)H dehydrogenase [quinone] 1 (NQO1), also known as DT-diaphorase, is an important player in the oxidative stress response [251,252]. NQO1 maintains the redox barrier between the organism and its environment, and is predominantly localized in the epithelial and endothelial tissues of mammalians [253]. NQO1 forms a homodimer and detoxifies ROS-generating quinones to hydroquinones. NQO1 follows a ping-pong mechanism. First, it uses NAD(P)H to reduce FAD and then catalyzes a two-electron reduction, regenerating FAD and yielding hydroquinone [254] (Figure 2).

Studies show that NQO1 is upregulated in certain types of cancer and associated with resistance towards anticancer drugs [255]. Furthermore, NQO1 polymorphism is associated with the development of certain cancer types [256,257]. These observations led to the interrogation of NQO1 as a cancer target. One notable example is the prodrug ß-lapachone (ß-Lap, ARQ 501), which consumes NAD(P)H and concomitantly generates O_2_^•−^; this was tested in numerous clinical trials including phase 2 but was not successful. Nonetheless, current studies are still exploring NQO1 as a direct target in cancer therapy [89,90]. Interestingly, recent findings suggest a novel approach of targeting NQO1 to modulate the TME in immunotherapy [258].

Besides NQO1, the human genome encodes the paralog NQO2, which is not as well studied as NQO1, and its role remains elusive [251,259]. NQO2 shows a different substrate specificity than NQO1, likely indicating a different biological role.

NQO2 is not affected by typical NQO1 inhibitors, such as dicoumarol, cibacron blue or phenindone, but is inhibited instead by the natural phenol resveratrol [91,259,260,261]. Current research focuses on furan-amidines as inhibitors of NQO2, but there is currently no selective inhibitor used in the clinic [93].

## 4. Exosomes in the Tumor Redox Microenvironment

Redox pathways govern fundamental physiological processes within the cell, but the redox landscape extends beyond the single cell—it is also crucial for multi-cellular systems, such as the tumor microenvironment (TME). In fact, the TME is characterized by oxygen depletion resulting in hypoxic conditions, which is associated with increased tumor aggressiveness and metastasis [262]. Hypoxia affects intercellular communication, for instance, by altering the release and uptake of extracellular vesicles, such as exosomes [263]. Several studies have shown that hypoxia-derived tumor exosomes are implicated in breast cancer [264], prostate cancer [265], pancreatic cancer [266], lung cancer [267], glioblastoma [268,269] and ovarian cancer [270]. In all of these instances, exosomes contribute to tumor growth, progression, and treatment resistance, which resulted in poor patient outcomes in some cases [271,272]. Therefore, current research seeks to understand the mechanisms behind exosomes in redox TME in order to improve current therapeutic strategies and develop novel ones, especially for the treatment of aggressive tumors. An overview of the role of exosomes in the redox TME is presented in Figure 3.

### 4.1. Redox Mechanisms of Tumor Exosomes

Exosomes are extracellular carriers that transport cytosolic biomolecules, such as miRNAs and proteins, from virtually all cells in the body to neighboring and distal cells via the endocytic pathway. Correspondingly, tumor cells generate distinct exosomes and other extracellular vesicles (EVs), such as microvesicles (MVs), to communicate and to invade other cells with their own tumorigenic-specific cargo. Thus, EVs are able to perpetuate and sustain the TME and modulate the redox environment [273,274,275].

Understanding the underlying mechanisms behind the regulation of exosome formation and release is an emerging field of research, which aims to reveal potential and novel cancer targets. The hypoxia-inducible factor (HIF) family are transcription factors that mediate expression of genes under hypoxic conditions, including genes that are associated with tumor growth and progression. According to several studies, HIFs are also involved in the formation and release of tumor-derived exosomes and other EVs in hypoxic conditions [263,266,268,276,277]. These studies showed that overexpression of HIFs correlates with increased release of tumor exosomes. Upon suppression and inhibition of HIF-1α and HIF-2α, exosome release levels reverted to those in normoxic conditions. Similar results were obtained via promotion of the negative regulators of HIF, i.e., prolyl hydroxylases (PHD1, PHD2, and PHD3). In addition, both HIF-1α and HIF-2α have been shown to bind to hypoxia response elements (HREs), which regulate a large array of genes that are also associated with hypoxia-derived exosomes. For instance, hypoxia in oral squamous cell carcinoma (OSCC) activates HREs in the promoter regions of exosomal microRNA-21 (miR-21), leading to miR-21 upregulation, which is linked to tumor growth [278]. Besides modulating exosomes, HIFs are a requirement in MV shedding of breast cancer cells [279]. These studies suggest that HIFs play an important and diverse role in the regulation of EVs, showing promise as drug targets in cancer therapy. In addition, redox pathways directly affect exosomal release via post-translational modification of exosomal surface proteins [280]. Specifically, redox-sensitive thiol groups can influence protein folding, acting as switches in the regulation of exosomal release [281]. 

Redox imbalance in the TME also alters the abundance of exosomal cargo proteins, and, subsequently, affects the redox states of cells that receive the exosomal cargo. For instance, the redox-sensitive signaling pathway PI3K/Akt/eNOS regulates the exosomal release of angpoietin-2 (Ang2), an important player in vascular remodeling of tumors [282,283]. Another example is the elevated exosomal release of a mutant SOD1 variant to neurons, which fosters disease spreading and has been described for motor neuron pathology in amyotrophic lateral sclerosis (ALS) [284]. Remarkably, exosomes also deliver increased levels of active HIF1-α and HIF2-α to healthy cells, transferring tumorigenic properties to the new host cell [285]. 

Redox imbalance also affects exosomal RNA cargo, which serves as a key mechanism in advancing tumorigenic stages [286,287]. An earlier study showed that exosomes laterally transfer RNA transcripts for CAT and SOD2, which promotes chemoresistance to pancreatic cancer cells; however, this study was conducted under very specific in vitro conditions, which are not necessarily physiologically relevant [288].

### 4.2. Leveraging Exosomes in Cancer Therapy

Exosomes are an important contributor to the redox TME, and hence have potential to be exploited in the development of cancer therapy. This is illustrated by a study on prostate cancer cells that showed inhibition of cancer cell growth upon treatment with exosome biogenesis inhibitors [289]. Current efforts in the development of novel therapeutic strategies against cancer also explore exosomal pathways that modulate the redox TME.

The HIF family, particularly the isoforms HIF1-α and HIF2-α, are important targets, especially in aggressive forms of cancer where drug resistance interferes with therapy [290,291,292]. Selective inhibitors of HIF1-α and HIF2-α, such as the compound 2ME2 NCD (panzem), showed promise in phase 2 clinical trials when used in combination with bevacizumab for carcinoid neuroendocrine tumor [94,293]. The first-in-class HIF2-α inhibitor—PT2385—and the more potent second-generation variant, PT2977, are both in phase 2 clinical trials [HIF-2 Alpha Inhibitor PT2385 in Treating Patients With Recurrent Glioblastoma (PT2385), NCT03216499; A Trial of PT2977 in Combination With Cabozantinib in Patients With Clear Cell Renal Cell Carcinoma (ccRCC), NCT03634540]. Besides HIF antagonists, researchers have also explored HIF regulatory pathways as pharmaceutical targets in order to enhance HIF degradation. One notable example is the known thioredoxin inhibitor Vorinostat (Zolinza), which inhibits HIF1-α and is approved for cancer treatment [294]. 

Interestingly, exosomes are also exploited as innovative drug delivery systems that target diseased cells with high selectivity [295,296]. Current promising developments in exosomal engineering include delivery of therapeutic cargo, e.g., the enzyme CAT, to neuronal cells in the treatment of Parkinson’s disease [297]. Another example is the delivery of miRNAs in the treatment of lung cancer [267]. Many challenges need to be addressed in developing exosomes as drug delivery systems for wide-scale clinical use, such as increasing the scale of production to meet expected market needs. However, exosomal biocompatibility and ability to incorporate many different therapeutically relevant payloads enable broad and well-adjusted application in cancer therapy.

## 5. Conclusions and Outlook

A growing body of research on the emerging field of redox biology illustrates the vast complexity of this discipline, and its implication for a variety of biological process. It is becoming clear that generalization of redox biology into oxidants and antioxidants, which are either “good” or “bad” for cancer, is difficult, if not impossible [9]. Nevertheless, our growing knowledge enables us to isolate specific pathways, enzyme kinetics, redox gradients and compartments as well as key players within a narrowly defined cellular environment, which can be exploited to modulate biological processes for disease management. It is well established that redox metabolism is involved in cancer development, progression and resistance to therapy. Specific and carefully adjusted intervention opens up the opportunity to tip the balance and disrupt the redox landscape in cancer cells. The ongoing advancements in cancer biology, for instance, the role of exosomal vesicles in the creation and maintenance of tumor microenvironments, will provide us with future targets and new therapeutic platforms. In addition, continuous developments in medicinal chemistry will provide us with novel tools, e.g. Nrf2–Keap1 modulators, to target key players in the redox landscape. Redox biology has the potential to bring novel therapeutic approaches and improve patient outcomes in the future.

## Figures and Tables

**Figure 1 cancers-12-01706-f001:**
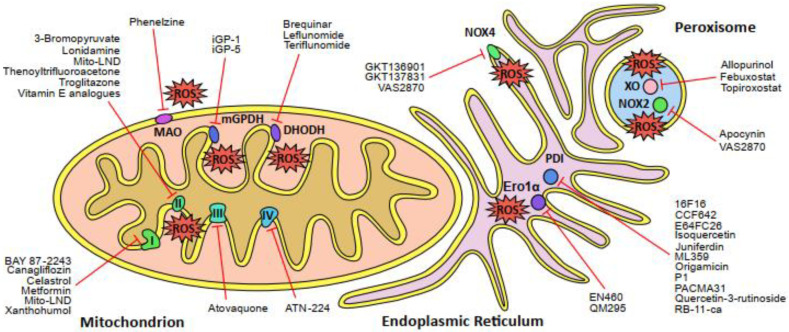
Schematic overview of the major sources of reactive oxygen species (ROS) in the cell and the corresponding inhibitors of those sites. The mitochondrial electron transport chain (ETC) complexes I, II and III generate ROS directly, whereas complex IV is the rate-determing step of the ETC. Other enzymes that produce ROS in the mitochondria are dihydroorotate dehydrogenase (DHODH), glycerol-3-phosphate dehydrogenase 2 (mGPDH or GPDH2) and monoamine oxidase (MAO). The endoplasmic reticulum (ER) comprises several sites of ROS production, such as NADPH oxidase 4 (NOX4) and the Ero1α-PDI oxidative protein folding pathway (Ero1α, ER oxidoreductin 1α; PDI, protein disulfide isomerase). Peroxisomes are another major source of cellular ROS production due to the activity of xanthine oxidase (XO) and NADPH oxidase 2 (NOX2). Names of pharmaceutical and experimental inhibitors are presented and the corresponding target sites of ROS production are indicated by red lines.

**Figure 2 cancers-12-01706-f002:**
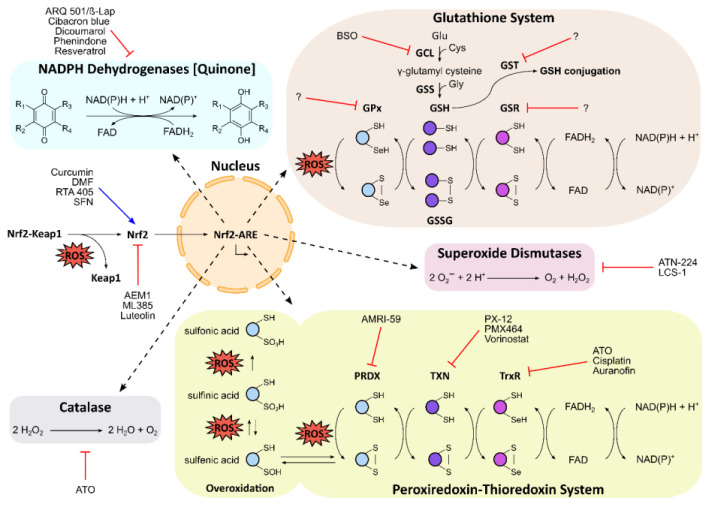
Schematic overview of the antioxidative landscape in the cell, and the corresponding modulators of key players. Reactive oxygen species (ROS) activate the Nrf2–Keap1 signalling pathway, resulting in induction of the antioxidant response elements (ARE) by Nrf2 in the nucleus. ARE comprise the glutathione (GSH) system, the peroxiredoxin–thioredoxin system, and antioxidative enzymes, such as NAD(P)H dehydrogenases [quinone], superoxide dismutases, and catalase. In the GSH system, the sequential activity of glutamate cysteine ligase (GCL) and glutathione synthetase (GSS) produces the tripeptide GSH. Glutathione peroxidases (GPx), which are often selenoproteins, use GSH to scavenge ROS, resulting in glutathione disulfide (GSSG). Glutathione reductase (GSR) regenerates GSH using FAD and NAD(P)H. GSH is also used for conjugation by glutathione S-transferase (GST) in cellular detoxification processes. In the peroxiredoxin–thioredoxin system, ROS are scavenged by peroxiredoxin (PRDX), resulting in oxidation of PRDX’s peroxidatic cysteine to sulfenic acid or disulfide bonds. Overoxidation yields sulfinic or irreversible sulfonic acid. Thioredoxin (TXN) and the selenoprotein thioredoxin reductase (TxR) regenerate PRDX using FAD and NAD(P)H. Names of pharmaceutical and experimental inhibitors and activators are presented, and the corresponding target sites are indicated by red lines or blue arrows, respectively.

**Figure 3 cancers-12-01706-f003:**
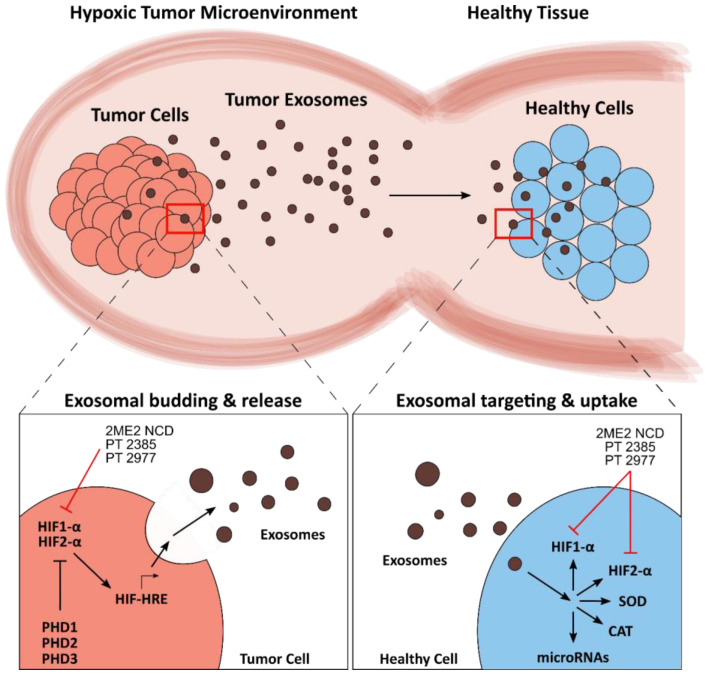
Schematic overview of exosomal modulation of the redox tumor microenvironment (TME). Tumor-derived exosomes maintain and propagate the TME by invading healthy cells (top). Prolyl hydroxylases (PHDs) negatively regulate hypoxia-inducible factors HIF1-α and HIF2-α. Hypoxic conditions activate HIFs, resulting in induction of hypoxia response elements (HREs) and an increase in exosome production (bottom left). The exosomal cargo contains active proteins, such as HIFs, but also microRNAs and RNA transcripts of redox proteins, such as superoxide dismutase (SOD) and catalase (CAT). Exosome uptake leads to the release of this cargo, which alters the redox landscape of the receiving cell (bottom right). The names of pharmaceutical and experimental inhibitors are presented, and the corresponding target sites are indicated by red lines.

**Table 1 cancers-12-01706-t001:** Overview of compounds presented in this review for targeting the redox landscape in cancer.

Redox System	Target	Compound	Application ^a^	Reference ^b^
Mitochondria, electron transport chain	Complex I	BAY 87-2243	various cancers	[24,25,26]
		Canagliflozin	various cancers (approved for type II diabetes)	[27,28]
		Celastrol	various cancers	[29]
		Metformin	various diseases	[30]
		Mito-LND	basic research	[31]
		Xanthohumol	various cancers	[32,33]
	Complex II	3-Bromopyruvate	various cancers	[34]
		Lonidamine	various cancers	[35,36]
		Mito-LND	basic research	[31]
		Thenoyltrifluoroacetone	basic research	[37]
		Troglitazone	basic research	[37]
		Vitamin E analogues (tocopherols & tocotrienols)	various cancers	[38,39]
	Complex III	Atovaquone	AML, NSCLC (approved for malaria)	[40]
	Complex IV	ATN-224	various cancers	[41,42]
Mitochondria, enzymes	DHODH	Brequinar	various cancers	[43,44]
		Leflunomide	various cancers (approved for rheumatoid arthritis)	[45]
		Teriflunomide	basic research (approved for multiple sclerosis)	[46,47,48]
	mGDPH (GDPH2)	iGP-1	basic research	[49]
		iGP-5	basic research	[49]
	MAO	Phenelzine	prostate cancer	[50,51]
ER	NOX1	GKT137831	basic research	[52,53]
	NOX4	GKT136901	idiopathic pulmonary fibrosis, type II diabetes, albuminuria	[53]
	Pan-NOX	VAS2870	basic research	[54]
	Ero1α	EN460	basic research	[55]
		QM295	basic research	[55]
	PDI	16F16	basic research	[56]
		CCF642	basic research	[57]
		E64FC26	basic research	[58]
		Isoquercetin	thrombus formation	[59]
		Juniferdin	basic research	[60]
		ML359	arterial thrombosis	[61]
		Origamicin	basic research	[62,63]
		P1	basic research	[64]
		PACMA31	basic research	[65]
		Quercetin-3-rutinoside	thrombus formation	[66]
		RB-11-ca	basic research	[67]
Peroxisomes	XO	Allopurinol	basic research (approved for hyperuricemia, gout)	[68]
		Febuxostat	basic research (approved for hyperuricemia, gout	[68]
		Topiroxostat	basic research (approved for hyperuricemia, gout	[68]
	NOX2	Apocynin	basic research	[69,70]
		VAS2870	basic research	[54]
Nrf2–Keap1 signaling pathway	inhibition of Nrf2	AEM1	NSCLC	[71]
		ML385	NSCLC	[71]
		Luteolin	NSCLC	[71]
	inhibition of Nrf2–Keap1 interaction (activation of Nrf2)	Curcumin	breast cancer	[72]
		Dimethyl fumarate	skin cancer, colon cancer (approved for multiple sclerosis, psoriasis)	[73,74,75]
		RTA 405	pancreatic cancer, lung cancer	[76,77]
		Sulforaphane	breast cancer, prostate cancer	[75,78]
Glutathione system	Glutamate cysteine ligase	Buthionine sulfoximine	MM	[79]
Peroxiredoxin–thioredoxin system	Peroxiredoxin	AMRI-59	NSCLC	[80,81]
	Thioredoxin	PX-12	various cancers	[82]
		PMX464	colorectal cancer	[83]
		Vorinostat	various cancers	[82]
	Thioredoxin reductase	Arsenic trioxide	AML, breast cancer	[82,84]
		Cisplatin	various cancers	[85]
		Auranofin	various cancers	[85,86]
Detoxifying enzymes	Catalase	Arsenic trioxide	HCC	[87]
	Superoxide dismutase 1	ATN-224	prostate cancer	[41]
		LCS-1	lung cancer	[88]
	NAD(P)H de-hydrogenase [quinone] 1	ARQ 501/ß-Lap	pancreatic cancer	[89,90]
		Dicoumarol	basic research	[91]
		Cibacron blue	basic research	[91]
		Phenindone	basic research	[91]
	NAD(P)H de-hydrogenase [quinone] 2	Resveratrol	basic research	[92]
		Furan-amidines	basic research	[93]
Redox tumor micro-environment	HIF1-α, HIF2-α	2ME2 NCD	various cancers	[94]
		PT 2385	RCC, glioblastoma	[94]
		PT 2977	RCC	[94]

^a^ classified as basic research unless advanced to clinical trials; ^b^ relevant articles mentioned in this manuscript; AML, acute myeloid leukemia; HCC, hepatocellular carcinoma; MM, multiple myeloma; NSCLC, non-small-cell lung carcinoma; RCC, renal cell carcinoma.

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
