# Peer review of "Targeting the Redox Landscape in Cancer Therapy"

_cancers, 2020, doi:10.3390/cancers12071706_

Round 1

Reviewer 1 Report

-The review focusses on the pathways involved in the production and inhibition of ROS and their inhibitors for cancer therapy. However, there is no discussion on how ROS contributes the cancer proliferation and metastasis. The authors may include a paragraph about how ROS contributes to different stages of cancer.
-The authors nicely represented the inhibitors involved in cellular oxidative pathways in the diagram. If possible authors may include a table with cellular oxidative pathways, their inhibitors, and the type of cancer affected with some important references.
-Line 157, change ca2+ to ca2+ (2+superscript).

-Reference 5 doi missing.

Author Response

* We would like to thank the reviewer for the helpful comments. We appreciate the generally positive perception of our work, and we are happy to incorporate the provided suggestions in order to improve our manuscript.

-The review focusses on the pathways involved in the production and inhibition of ROS and their inhibitors for cancer therapy. However, there is no discussion on how ROS contributes the cancer proliferation and metastasis. The authors may include a paragraph about how ROS contributes to different stages of cancer.

* The reviewer raised an important aspect that we should address in our manuscript. We added several sentences and references to the introduction (line 35 – 50), in which we state how ROS contributes to cancer proliferation and metastasis, and describe the different stages of cancer development and the associated contribution of the redox landscape. Furthermore, we added additional lines to explain the purpose of the different redox-associated cancer therapies to the introduction (line 55 – 59)
-The authors nicely represented the inhibitors involved in cellular oxidative pathways in the diagram. If possible authors may include a table with cellular oxidative pathways, their inhibitors, and the type of cancer affected with some important references.

* We agree with the reviewer that a table summarizing all compounds/inhibitors of this review and the corresponding targets would be highly beneficial for the reader. We have added this table, including several additional references, at the end of the introduction, and made related changes to the manuscript accordingly.

-Line 157, change ca2+ to ca2+ (2+superscript).

* We changed this term accordingly.

-Reference 5 doi missing.

* The reviewer is right, and we were not able to retrieve a DOI for this particular reference. Therefore, we replaced this reference (PMID: 2400824) with a more current one (PMID: 20370557), which provides a more comprehensive overview of the stated topic. We also checked our references, and with a few exceptions, they all possess a doi.

Reviewer 2 Report

Narayanan et al. present an overview of ROS metabolism, antioxidants, and how drugs targeting these compounds can be employed in cancer therapy. Overall, this is a well written review that is easy to follow, introduces the fundamental biochemistry behind these redox systems, and frames the discussion with respect to known drugs and their targets.

Main comments:

  1. It could be interesting to have a brief discussion on tumor heterogeneity, not only at the TME level but also considering patient variability. How do the authors view application of these drugs and therapy strategies in different patient subgroups?
  2. Substantial redox alterations, including in expression of antioxidant enzymes, may occur with progression or be associated with overall tumor heterogeneity (e.g. PMID 30606699). Further, different antioxidant enzymes have very distinct kinetics, and may lead to substantial intracellular ROS gradients (PMID 28279943), which should be even more pronounced in the TME due to metabolic gradients (PMID 28246332). Yet, as formulated by the authors, it appears that targeting any of the antioxidants would be equally viable in cancer therapy. It would be interesting if the authors could comment on these issues.
  3. Do all compartments equally contribute for the intracellular production of ROS?
  4. Would immuno-modulatory approaches be helpful when synergized with redox-driven strategies?
  5. Sulfiredoxin inhibitors have been observed in vitro. Would those be suitable candidate therapies?

Minor:

  1. Line 361-363: Prxs are known for their redox signaling properties, so it is perhaps worth to revise their “most important function”.
  2. Lines 528-529: it is perhaps better to be cautious regarding the lateral transfer, as the cited study was applied in vitro in very specific conditions that not necessarily translate to the in vivo
  3. Reference missing for the sentence in lines 416-418.
  4. Figure 2: “ROS“ may be replaced by the specific species that reacts chemically (e.g. all of Prx metabolism)
  5. Please fix H2O2 and Ca2+ throughout
  6. Typos: lines 70, 270, 390
  7. Something strange going on with lines 541 – 544.

Author Response

* We are glad that the reviewer recognizes the value of our review. Furthermore, we greatly appreciate the extensive and helpful comments from the reviewer. We have addressed the following concerns and we will be happy to clarify any other concerns the reviewer may have on our manuscript.

Narayanan et al. present an overview of ROS metabolism, antioxidants, and how drugs targeting these compounds can be employed in cancer therapy. Overall, this is a well written review that is easy to follow, introduces the fundamental biochemistry behind these redox systems, and frames the discussion with respect to known drugs and their targets.

Main comments:

  1. It could be interesting to have a brief discussion on tumor heterogeneity, not only at the TME level but also considering patient variability. How do the authors view application of these drugs and therapy strategies in different patient subgroups?

* The reviewer raises an interesting question. As mentioned above, we elaborated on the heterogeneity of tumor cells regarding type, tissue and stage, but we do not extensively discuss the impact of genetic variations in patient subgroups. We assume that genetic variation is an important contributor to therapy outcome. For instance, we mention that NQO1 polymorphism is associated with certain cancer types (line 484 – 485). Nevertheless, we think that a detailed discussion on this particular issue is beyond the scope of this review, which addresses the redox landscape from a broader perspective. In any case, we added three sentences (line 55 – 59), in which we mention different types of approaches and their aim in cancer therapy, as well as the role of genetic polymorphism in the patient.

  1. Substantial redox alterations, including in expression of antioxidant enzymes, may occur with progression or be associated with overall tumor heterogeneity (e.g. PMID 30606699). Further, different antioxidant enzymes have very distinct kinetics, and may lead to substantial intracellular ROS gradients (PMID 28279943), which should be even more pronounced in the TME due to metabolic gradients (PMID 28246332). Yet, as formulated by the authors, it appears that targeting any of the antioxidants would be equally viable in cancer therapy. It would be interesting if the authors could comment on these issues.

* we agree with the comments of the reviewer that the redox landscape and the underlying details, including heterogeneity of tumors, enzyme kinetics, metabolic gradients etc. are highly complex.

We attempt to address these issues for instance in the section on Nrf2-Keap1, where we describe the different approaches on inhibitors and activators, indicating that targeting redox-related proteins is a doubled-edge sword involving many factors, including cancer type, stage and tissue. Furthermore, our overview of the different pharmaceutical approaches mentions that some approaches are more successful than others, indicating that simply targeting one single redox enzyme is not sufficient. In our outlook (line 588 – 590), we highlight that there is no ‘general’ oxidant or antioxidant, and that there are many additional aspects to consider for cancer therapy. We also included “enzyme kinetics” and “redox gradients” as per  the comment of the reviewer.

  1. Do all compartments equally contribute for the intracellular production of ROS?

* The reviewer raises an important point. There are studies indicating that the largest absolute amount of ROS is produced in peroxisomes but mitochondria also contribute substantially to the overall ROS production. Other studies point to the relative amount of ROS production, which is predominantly attributed to the ER. We added a few supplementary sentences and the corresponding references to explain this fact (see line 73 – 78). In line with this, we moved the second half of the sentence from line 185 – 186 to 77 – 78, which previously addressed the role of the ER in this context.

  1. Would immuno-modulatory approaches be helpful when synergized with redox-driven strategies?

* This is an interesting and important aspect in cancer therapy; however, we feel that an extensive elaboration is beyond the scope of this review. We, however, added a statement (line 58 – 59) mentioning that redox-driven strategies are explored in immune-modulatory approaches. Furthermore, we would like to point to the section on NQO1, where we state a current approach targeting NQO1 to modulate immunotherapy (line 489 – 490).

  1. Sulfiredoxin inhibitors have been observed in vitro. Would those be suitable candidate therapies?

* We would like to thank the reviewer for mentioning the sulfiredoxin to us, an important antioxidative enzyme. Indeed, it is an interesting enzyme that we initially did not mention but we have now added some information about the sulfiredoxin in the manuscript (line 397 – 399). Since sulfiredoxin is a relatively new development that has not been studied as extensively as other redox enzymes, and given the length limitation of this manuscript, we think that a detailed elaboration is beyond the scope of this review; however, we reference current reviews and a recent study on sulfiredoxin inhibitors. It is premature but given the success of thioredoxin inhibitors in cancer therapy (cisplatin, ATO), we think that sulfiredoxin could be a similar successful drug candidate. 

Minor:

  1. Line 361-363: Prxs are known for their redox signaling properties, so it is perhaps worth to revise their “most important function”.

* We changed this phrase to “they play an important role in”.

  1. Lines 528-529: it is perhaps better to be cautious regarding the lateral transfer, as the cited study was applied in vitro in very specific conditions that not necessarily translate to the in vivo

* We added the following sentence to address the limitation of this study: “[…]; however, this study was conducted under very specific in vitro conditions, which are not necessarily physiological relevant”

  1. Reference missing for the sentence in lines 416-418.

* We admit that this section might be somewhat misleading. We were referring to the comprehensive review from Che at al (PMID: 26475962). In order to clarify the source of our statements, we added the following to lines 444 – 445: “[…], as summarized in the review by Che et al. [233]. Che et al. describe that […]”

  1. Figure 2: “ROS“ may be replaced by the specific species that reacts chemically (e.g. all of Prx metabolism)

* We thank the reviewer for this suggestion. Even though there might be a dominant version of a specific ROS that reacts with a particular protein/enzyme, there are instances, in which another ROS can react and induce a similar response. An extensive list would be beyond the scope and limiting to only a major ROS would be misleading; hence, we argue that the generic “ROS” symbol in figure 2 is the best compromise.

  1. Please fix H2O2 and Ca2+ throughout

* We changed this term accordingly.

  1. Typos: lines 70, 270, 390

* We corrected the words ‘activity’, ‘catalase’ and ‘polyphenolic’.

  1. Something strange going on with lines 541 – 544.

* We agree with the reviewer. We changed the font accordingly.

Reviewer 3 Report

The manuscript "Targeting the redox landscape in cancer therapy" is well written and systematically and clearly describes the progress in the emerging field of redox-oriented cancer therapy.

However, several corrections should be done before acceptance of the manuscript for publication:

line 57: I suggest to change the sentence to: giving rise to more stable hydrogen peroxide (H2O2) but also to more toxic ROS, e.g. hydroxyl radical (OH).

line 310: the term "non-covalent" is misleading in this context. I suggest to use: "non-covalently interacting small molecules".

line 401: I suggest to use the term: "histone deacetylase inhibitor suberoylanilide"

line 425: the word "faired" should be changed to "failed"

Multiple cases throughout the text: superoxide is O2⋅-, hydrogen peroxide is H2O2

Author Response

* We appreciate the helpful comments and suggestions from the reviewer. We addressed all points in order to improve our manuscript.

The manuscript "Targeting the redox landscape in cancer therapy" is well written and systematically and clearly describes the progress in the emerging field of redox-oriented cancer therapy.

However, several corrections should be done before acceptance of the manuscript for publication:

line 57: I suggest to change the sentence to: giving rise to more stable hydrogen peroxide (H2O2) but also to more toxic ROS, e.g. hydroxyl radical (OH).

* We agree with the suggestion of the reviewer and we have made the corresponding changes.

line 310: the term "non-covalent" is misleading in this context. I suggest to use: "non-covalently interacting small molecules".

* We replaced our wording as per the suggestion of the reviewer.

line 401: I suggest to use the term: "histone deacetylase inhibitor suberoylanilide"

* We changed the sentence to the following one in order to accommodate the suggestion from the reviewer: “[…] The third compound is the histone deacetylase inhibitor suberoylanilide, commonly known as Vorinostat (Zolinza). […]”.This change also led to the removal of the abbreviation SAHA from the manuscript. Consequently, we had to replace figure 2 to accommodate this change.

line 425: the word "faired" should be changed to "failed"

* We could not replace “faired” by the word “fail” without restructuring the entire sentence. Therefore, we replaced the word “faired” with the phrase “did not match”, which aligns better with the rest of the sentence.

Multiple cases throughout the text: superoxide is O2-, hydrogen peroxide is H2O2

* We corrected this throughout the manuscript.